# Change for the Better: Severe Pneumonia at the Emergency Department

**DOI:** 10.3390/pathogens11070779

**Published:** 2022-07-08

**Authors:** Dariusz Kawecki, Anna Majewska, Jarosław Czerwinski

**Affiliations:** 1Department of Emergency, Medical University of Warsaw, 02-005 Warsaw, Poland; dkawecki@wum.edu.pl (D.K.); jczerwinski@wum.edu.pl (J.C.); 2Department of Medical Microbiology, Medical University of Warsaw, 02-004 Warsaw, Poland

**Keywords:** aging, community-acquired pneumonia, diagnosis, emergency department, microbiology examination, mortality, PEDRO, pneumonia

## Abstract

This is a single-centre observational study of adult patients with severe pneumonia requiring hospitalization conducted at the emergency department. During the observation period (94 weeks), 398 patients were diagnosed with severe pneumonia and required further treatment at the hospital. The median age of patients was 73 years. About 65% of patients had at least one chronic comorbidity. Almost 30% of patients had cardiovascular disorders, and 13% had diabetes mellitus. The average Emergency Department length of stay was 3.56 days. The average length of hospitalization was 15.8 days. Overall, 94% of patients treated for pneumonia received a beta-lactam antibiotic. The median time from ED admission to the administration of the first dose of antimicrobial agent was less than 6 h. Microbiology test samples were obtained from 48.7% patients. Gram-positive cocci were isolated most commonly (52.9%) from blood samples. Biological material from the lower respiratory tract was collected from 8.3% of patients, and from 47.2% of positive samples, fungi were cultured. The urine samples were obtained from 35.9% patients, and Gram-negative rods (76%) were isolated most commonly. Overall, 16.1% of patients died during the hospitalization. The mean age of patients who died was 79 years. This observational study is the first single-centre study conducted as part of the Polish Emergency Department Research Organization (PEDRO) project. It aims to provide up-to-date information about patients with pneumonia in order to improve medical care and develop local diagnostic and therapeutic recommendations.

## 1. Introduction

### 1.1. Background

The emergency department (ED) plays a pivotal role in providing the community access to acute health care. Because of the unscheduled and episodic nature of health emergencies, clinical decision making by ED staff is a real challenge and a complex process. Every patient who comes through the door is an unknown and may suddenly develop worsening symptoms. Clinicians at the ED are obligated to handle plenty of various issues simultaneously, most of which cannot be foreseen in advance and optimized to “exist at the edge of chaos”. In order to ensure proper care of patients, it is necessary to quickly answer some questions: what is the likely diagnosis, what is the correct diagnostic pathway, what medication should be administered, and which unit should the patient be transferred to [1].

The goal of EDs is to provide evidence-based, high-quality, intensive, short-term observation and optimal antibiotic regimens for selected ED patients and reduce unnecessary hospital admissions and/or premature discharges due to over- and underdiagnosis [2]. Pneumonia remains a challenge to the clinician at the emergency department. It is a common cause of ED visits, including visits of elderly patients, whose pneumonia can be difficult to diagnose due to the non-specificity of symptoms and other comorbidities [3]. The experience of researchers from other countries shows that pneumonia is a common infection and causes significant morbidity and mortality [4,5,6,7]. No complex, up-to-date studies among Polish patients with pneumonia have been conducted so far. We do not know the real scale of the problem in Poland, characteristics of patients, and aetiological factors of infection [8].

The national recommendations for the Management of a patient with suspected severe infection at the Hospital Emergency Department, published in 2014, state that there are no Polish data describing the management of a patient with symptoms of severe infection at the ED [9]. Despite the fact that almost 8 years have passed since the communication of the need to obtain and compile the data, we still do not have any national experience and recommendations regarding the management of a patient in severe condition, including pneumonia requiring hospitalization, at EDs. The authors of the study refer to experiences of other researchers that were published between 1990 and 2006. Since then, there have been many changes, including but not limited to demographic conditions (e.g., aging of the population, high percentage of immunosuppressed people in the population), aetiological factors of infections (new and re-emerging infections), microbial antibiotic susceptibility profiles, diagnostic methods, and cost policy in hospitals; therefore, there is a need to characterize patients diagnosed at the ED in order to improve patient care [9]. It should be noted that Poland compares very unfavourably in terms of the pneumonia death rate to Europe. According to the Eurostat estimate, 131,450 people died from pneumonia in the European Union countries (EU-28) in 2016. The standardised death rate (SDR; three-year average) from pneumonia stood at 27 deaths per 100,000 EU inhabitants in 2017 (Figure 1). The SDR for the EU population aged ≥ 65 years was nearly 5-fold higher (126.52/100,000). In Poland, very high SDRs have been reported for years, with over 55 deaths per 100,000 inhabitants in some voivodeships [10].

### 1.2. Objective

The objective of this study is to evaluate demographic features, clinical patterns, and history of hospitalisation associated with patients who suffer from pneumonia requiring hospitalization and their impact on mortality.

The analysis of clinical records of adult patients admitted to the ED between 1 January 2019 and 15 October 2020 is a starting point for organized activities with the participation of other emergency departments in Poland to understand the characteristics and impact of this disease, improve the quality of care, and, above all, reduce mortality rates due to pneumonia in Poland.

## 2. Methods

This research was designed as a single-centre observational study of adult patients with severe pneumonia requiring hospitalization. The study was conducted at the Emergency Department of Clinical Hospital located in the centre of Warsaw, Poland. It is the first study conducted as part of the PEDRO (Polish Emergency Department Research Organization) project.

We reviewed and compared clinical records of patients who were admitted to the ED between 1 January 2019 and 15 October 2020; after 15 October 2020 only patients with confirmed COVID-19 were admitted to the hospital’s ED; therefore, the study was terminated. The data set for the analysis included only patients with first-listed diagnosis of pneumonia, meaning that pneumonia was the main reason for hospitalisation.

The key inclusion criteria were patients over 18 years of age with signs and symptoms of pneumonia who needed hospitalisation. The diagnosis of pneumonia was suspected based on the American Thoracic Society criteria, i.e., when:-The patient reported at least one of the following signs: cough, fever or chills, difficulty, breathing, chest pain, low energy and poor appetite, nausea, and diarrhoea;-Reduced or abnormal sounds were heard in the lungs during the physical examination;-Radiological (chest X-ray/computed tomography scan, CT) evidence of (an) infiltrate(s) consistent with pneumonia were observed [11].

The assessment of indications for hospitalization was conducted in accordance with the national recommendations stating that a decision whether to refer a patient to the hospital or not should take into consideration the patient safety and economic aspects. Therefore, indications for hospitalization result from the severity of the patient’s condition, presence of comorbidities that may contribute to the unfavourable course of infection, ability to take oral medications, and social conditions of the patient [12].

We considered patients with lung disease that was not related to pulmonary infection and COVID-19 patients (people infected with SARS-CoV-2 were being transferred to a dedicated COVID-19 hospital) to be exclusion criteria in this study.

For each patient, information about demographics (age and sex), mortality, ED and hospital length of stay (LOS), clinical data (based on triage, signs and symptoms, and medical history), microbiological test results, and radiological features was collected. All data were entered in the questionnaire and checked by two persons (double checked).

Microbiology testing was conducted at the hospital laboratory. A decision to perform microbiological testing was made on the basis of the national Recommendations for the management of community-acquired respiratory tract infections [12], in accordance with the recommendations:The performance of microbiology testing, in particular a sputum culture, should be considered when the presence of a risk factor for infection with multidrug-resistant microorganism is identified, or aetiology of infection may be different than the most common one;Sputum culture was performed in patients hospitalized for moderate or severe community-acquired pneumonia who expectorated purulent secretion; sputum cultures are recommended prior to initiating antibiotic therapy. Peripheral blood culture can also be performed in those patients;In the event of severe pneumonia, in particular when the history shows lack of response to therapy with beta-lactam antibiotics, the determination of *Streptococcus pneumoniae* and *Legionella pneumophila* antigens in the urine is recommended [12].

Microbiology test results were interpreted in accordance with the principles specified in the Indications for the performance of microbiological testing in hospitalized patients [13] and the ATS/IDSA Clinical Guideline on Community-Acquired Pneumonia [14].

### Statistical Analysis

In descriptive statistics, means, medians, standard deviations, and extreme values were used for continuous variables, while counts and proportions for categorical variables. Goodness-of-fit χ^2^ tests were applied to check if categorical variables follow uniform distribution. To compare categorical variables between groups, Pearson’s χ^2^ tests for independence were used. For comparisons of continuous variables between two groups, Mann–Whitney tests were applied. Survival times were compared with log-rank tests. All statistical hypotheses were verified at 0.05 level of significance. Calculations were performed using IBM SPSS Statistics ver. 20.0.

## 3. Results

Between 1 January 2019 and 15 October 2020 (94 weeks, 21.5 months), 61,108 adults presented at the ED of the University Hospital located in the centre of Warsaw (according to the Central Statistical Office of Poland (Polish abbreviation: GUS), the number of inhabitants was 1790.6 thousand and 1794.2 thousand in 2020 and 2021, respectively) [15]. In total, 8022 (13,1%) people were admitted to the hospital for further treatment; i.e., the average weekly number of patients admitted to hospital units following the ED visit was 85.3. During the observation period, 398 patients were diagnosed with severe pneumonia and required further treatment at the hospital, which was 5% of all admissions to our hospital during the 94-week observation period.

### 3.1. Demographic Characteristics and in-Hospital History of Patients Visiting the Emergency Department Who Were Diagnosed with Pneumonia during the 94-Week Observation Period

The median age of patients diagnosed with severe pneumonia at the ED was 73 years. The ratio of admitted women to men was 1:1.7. (a statistically significant difference, *p* < 0.001). The analysis of incidence of pneumonia requiring hospitalization across age groups 18–65, 66–80, and >80 years did not reveal any statistically significant differences; the ratio of hospitalized patients was 1.07:1:1.06. In total, 278 patients (70.2%) were brought to the hospital by the emergency medical team (EMT).

Forty-five (11.3%) patients were referred for further treatment at the intensive care unit (IUT) from the ED due to a diagnosis of severe pneumonia. The majority of patients (340; 85.4%) continued their therapy at the internal medicine unit. Sixty-four (16.1%) patients died during the hospitalization. Demographic characteristics, history of patients visiting the ED who were diagnosed with pneumonia, and intra-hospital movement and in-hospital mortality rates are shown in Table 1.

Triage—the emergency medical segregation that enables to select patients who need immediate care—has been recommended in Poland since 2019. Triage is based on the assessment of vital signs and is a part of the Mode of Patient Care at the Emergency Department (Polish abbreviation: TOPSOR) [16].

Initial patient assessment includes measurements of body temperature (T), blood pressure (systolic; SP/diastolic; DP), heart rate (HR), oxygen saturation (OS), respiratory rate, glycaemia, body weight, assessment of consciousness level using the Glasgow Coma Scale (GCS) or Alert Verbal Painful Unresponsiveness (AVPU) scale, assessment of pain severity, and performance of an electrocardiogram (ECG). The assessment of the patient health condition at hospital admission (TRIAGE) was performed in the majority of patients (between 306 and 362, depending on the assessed parameter). The triage data (those recorded in medical records) are presented in Table 2. The patient stratification at the ED did not take in account the severity of community-acquired pneumonia assessed using CURB, CURB-65, Pneumonia Severity Index for (PSI) for community-acquired pneumonia (CAP), and parameters included in Severe Community-Acquired Pneumonia (SCAP).

### 3.2. Main Presentation Symptoms, Clinical Diagnosis, and Co-Occurring/Coexisting Conditions

Symptoms reported most commonly by patients presenting at the ED included dyspnoea, malaise, and fever/feeling of fever. The majority of patients (58.0%) presented only one symptom. The presence of two symptoms was reported by 24.9% of patients. Around 65% of patients with pneumonia diagnosed at the ED had at least one chronic comorbidity. Almost 30% of patients had cardiovascular disorders, and 13% of patients suffered from diabetes mellitus. Aspiration pneumonia was diagnosed in 16 subjects (4%): patients with disorders of consciousness, in advanced age, with alcohol dependence, or as a complication following gastroscopy or gastric lavage after attempted suicide. Detailed characteristics are presented in Table 3.

### 3.3. Length of Patient Treatment at the ED and Overall Hospitalization Time

The average ED length of stay of patients diagnosed with pneumonia was 3.56 days (median: 1.88 days). The average length of hospitalization was 15.8 days (median: 12 days). Detailed information is provided in Table 4.

### 3.4. Antimicrobial Therapy, Imaging Examinations, and Microbiological Test Results

The majority of patients (347; 95%) received an antimicrobial agent during the ED stay. In 18 (5%) patients, the therapy was initiated after the transfer to the unit. Overall, 68% of patients received an antibiotic within ≤1 h of being admitted to the ED. The median time from ED admission to the administration of the first dose of antimicrobial agent was less than 6 h (Table 5).

In the course of hospitalization, 94% of patients treated for pneumonia received a beta-lactam antibiotic. Ceftriaxone, belonging to third-generation cephalosporin, was the most commonly administered medication (77.5% of patients). The main indications for ceftriaxone administration include both community-acquired and hospital-acquired pneumonia as well as bacteraemia associated with those infections. The empirical treatment was in line with the national recommendations and the ATS/IDSA Clinical Guidelines on the Management of Community Acquired Pneumonia [12,14]. The combination therapy with two antibiotics was used at the ED in 108 (31%) patients (72; 42% in 2019 and 36; 14% in 2020). Two (0.6%) subjects received three antimicrobial agents. Meropenem was administered to colonised patients or those infected with multidrug resistant (MDR) bacteria. The antiviral agent (ganciclovir) was ordered to a patient with transplanted organ who was suspected to have CMV pneumonia. Oseltamivir was administered in the case of confirmed or suspected influenza, while pentamidine was administered to one OLTX and LTX patient after CMV and SARS-CoV-2 infections were excluded and with suspected *Pneumocystis jirovecii* infection. The patient was allergic to sulfamethoxazole + trimethoprim. In eight patients, infections with other fungi were suspected, and antifungal agents (fluconazole, caspofungin) were administered. Antimicrobial treatments of inpatients are shown in Table 6.

#### 3.4.1. Imaging Examinations

The imaging examination result was one of diagnostic criteria of pneumonia. Chest X-ray or chest CT scan was performed in patients, with chest X-ray done more often (93% of patients) during the first year of observation (2019) and chest X-ray and CT performed in the case of doubts (38%). In the subsequent year of observation (2020), chest CT was ordered much more often (74,8%) (statistically significant, *p* < 0.001). (Table 7). It was related to the epidemic situation and SARS-CoV-2 infections. Chest CT offers advantage over chest X-ray in detecting lesions in the lungs in COVID-19 patients.

#### 3.4.2. Microbiological Examination

Microbiology test samples were obtained from 194 (48.7%) patients. No bacteria were cultured, and no fungal antigens were detected from 69.6% of samples.

#### 3.4.3. Blood Microbiological Testing

Positive results were produced for 30.4% of blood samples, with bacteria grown THAT were considered contamination in 3.1% of cases (in accordance with the clinical and laboratory criteria recommended by the Centers for Disease Control and Prevention (CDC) and the third international consensus definitions of sepsis and septic shock) [17]. In total, 51 microbial isolates were cultured from samples considered to be true-positive (27.3%). Gram-positive cocci (52.9%) were isolated most commonly, of which coagulase-negative staphylococci (51.8%) made up the largest group. *Candida* spp. antigens were detected in 11.8% of blood sample tests. The species of detected microorganisms are presented in Table 8.

#### 3.4.4. Microbiological Testing of Material from the Lower Respiratory Tract

Biological material from the lower respiratory tract was collected from 33 (8.3%) patients.

The tracheal aspirate (20/33; 60.6%) was collected most frequently, while the sputum (6/33; 18.2%), bronchoalveolar lavage fluid (BALF) (3/33; 9.1%), and pleural fluid (4/33; 12.1%) were obtained much less often. Bacteria and fungi were cultured from 25 (75.7%) respiratory tract samples. No significant numbers of microorganisms were found in 12.1% (4/33) of samples (negative result). Bacteria grown were considered specimen contamination in four (12.1%) cases. Fungi were cultured from 47.2% of samples. The species of detected microorganisms are presented in Table 8.

#### 3.4.5. Urine Microbiological Testing

The urine for microbiological testing was obtained from 143 (35.9%) patients. No microorganisms were cultured from 62.2% of urine samples (negative result). The urine sample contamination requiring repeated test occurred in 9.1% of cases. Significant numbers of microorganisms indicating a positive test result were found in samples obtained from 41 (28.7%) patients. A total of 13 microbial species were cultured. Gram-negative rods (76%) were isolated most commonly. The species of detected microorganisms are presented in Table 8.

### 3.5. Characteristics of Mortality after ED Admission with Recognized Severe Pneumonia

Among patients with severe pneumonia who were admitted to the hospital, 64 people died. As many as 26 (40.6%) patients died during ≤7 days of hospitalization. One person died during the ED stay, and 23 people (23/64; 35.9%) died during therapy at the ICU, which was 51% of patients continuing their therapy at the ICU. Survival times were significantly shorter in patients transferred from the ED to the ICU as compared to those hospitalized at other units (*p* < 0.001). The mean age of those who died was 79 years with the median of 81 years, and it was not significantly different in subsequent years (mean age: 80.5 years; median: 81 years in 2019; mean age: 77.8 years; median: 81 years in 2020). Thirty-four men (mean age: 77.3 years; median: 74.5) and thirty women (mean age: 80.7; median: 84.5) died. The median survival of hospitalized patients was 9 and 12 days (range: 1 to 60 days; mean: 11.7 and 13.6) in 2019 and 2020, respectively. The following were regarded to be major causes of death by ICD-10 codes: cardiac arrest, cause unspecified (I46.9): 23, 35.9%; shock, unspecified (R57): 14, 21.9%; heart failure, unspecified (I50.9): 8, 12.5%; and respiratory arrest (R09.2): 7, 10.9%. The majority of those who died at the hospital were brought to the ED by the medical transport team (92.1%; *p* < 0.001).

It was noted that the most common symptoms at admission to the ED (dyspnoea, malaise, and fever/feeling of fever) were not reported by people who died or their families; therefore, they did not have any statistically significant effect on the prognosis as the incidence proportion of death. Similarly, the most common comorbidities (listed in Table 3) did not significantly affect the incidence proportion of death. The statistical analysis showed that only oncological history, solid organ transplantation, and mental disorders increased the risk of death as independent factors (Table 9).

Patients received the antibiotic(s) in accordance with the applicable guidelines. Antimicrobial therapy was applied in 87.5% (56/64) of patients. It shows that the time of administration ≤ 3.20 h and ≤6 h from ED admission does not significantly affect the incidence proportion of death (*p*-values were 0.301 and 0.117, respectively). It has been demonstrated that the survival time was significantly longer when the antibiotic was administered no later than 12 h after patient admission (*p* = 0.043).

## 4. Discussion

The clinical presentation of pneumonia varies, ranging from mild pneumonia characterized by fever and productive cough to severe pneumonia that is characterized by respiratory distress and requires hospitalization [18,19]. There is a common view that clinical diagnosis of pneumonia does not present any problem if the symptoms are classic. In the presented study, nearly 66% of patients were aged more than 65 years. The symptoms of pneumonia in the elderly can be subtle, and fever is not a dominant symptom [20,21].

During the observation carried out, patients diagnosed with severe pneumonia most frequently reported the presence of dyspnoea (42.7%) and malaise (31.4%). Fever occurred in 20.6% of patients and cough in 7% of patients only. One symptom was reported by as many as 58% of patients, and two symptoms were present in 24.9% of subjects. In view of the above, subtle clinical symptoms of CAP should be proactively sought in very elderly people admitted to the ED due to unexplained falls, urinary incontinence, dementia, or sudden exacerbation of any comorbidity (e.g., diabetes mellitus, congestive heart failure, Parkinson’s disease) [20].

Pneumonia remains an important cause of death in the elderly population aged over 65 years. The older adults have higher rates of hospitalization and are more likely to die as a result of pneumonia [22,23]. The causes are complex and include the presence of multiple comorbidities, changes in basic lung physiology, changes in the immune system, and upper airway colonization with virulent organisms, which predisposes to the occurrence of silent aspiration/micro-aspiration of bacteria and fungi from the upper respiratory and digestive tracts [20,23,24].

According to the data published by the Central Statistical Office of Poland (GUS) with regard to the size and structure of the population in Poland between 1989 and 2019, the number of people aged over 65 years increased by 25.3% in the reported period [25].The median age of patients included in our study was 73 years, with people aged over 65 years accounting for over 65% of the cohort. This forces the adaptation of the national health care system to the changing Polish population, especially since pneumonia remains one of the most important causes of mortality in Poland [10].

Pneumonia is a leading cause of hospitalization. Approximately one million people hospitalized annually across the European Union require hospital treatment, and the need for hospitalization in adults increases with patient age [26].

During the 94-week observation at the ED, pneumonia was an indication for hospitalization in 5% of patients admitted to the hospital. Among adults aged ≥ 50 years, the rate of CAP-related hospitalizations in Poland in the period between 2007 and 2009 was 363.9/100,000 person-years. For comparison, in neighbouring countries of Poland, CAP-related hospitalization rates were 456.6/100,000 (Czech Republic) and 504.6/100,000 (Slovakia) in 2010 [27]. It has been demonstrated that the hospitalized rate among adults aged ≥ 65 years in the U.S. was 18.3% (between 847 and 3500 per 100,000) of community-acquired pneumonia cases and that adults aged over 90 years were more than five times more likely to be hospitalized than individuals aged 65–69 years in the pre-COVID-19 era [4,28].

In Poland (during the 7-year observation; 2009–2016), the number of community-acquired pneumonia cases varied between 5.5 and 6.6 occurrences per 100,000 population among patients aged ≥ 60 years [29].

In the study conducted by Kaplan and co-workers, the death rate among inpatients with confirmed community-acquired pneumonia was nearly 11%, and the mortality doubled with age: from 7.8% among patients aged 65–69 years to 15.4% among individuals aged 90 years and older [4]. It is not surprising that mortality rates in the elderly population with pneumonia are higher than in younger adults. In patients aged over 65 years, mortality rates ranged from 10% up to 30% in those managed at the ICU [4,5,6,22].

In our observation, the death rate among hospitalized patients was 16%; the mean age of those who died was 79 years (median: 81 years). The median ages of women and men who died were not statistically different; they were 84.5 and 74.5 years (*p* = 0.557); 36% of deaths occurred at the ICU. In the study conducted by Tichopad et al., the average fatality rate for all adults aged ≥ 50 years was 18.6% in Poland (2007–2010) and in the neighbouring countries, 21.7% (Czech Republic) and 20.9% (Slovakia) in 2010, and the incidence, fatality, and likelihood of hospitalization increased with advancing age [25].

The identification of individuals with severe infection requiring hospitalization or ICU admission is helpful in the stratification of patients. In many countries, the Pneumonia Severity Index (PSI), parameters included in the Severe Community-Acquired Pneumonia (SCAP), and CURB-65 are widely used and helpful [14,20]. The application of those criteria may have an impact on the ED waiting time of a patient and a decision regarding the place of further treatment. The avoidance of delays in consultation can affect patient flow and contribute to prolonged lengths of stay (LOS) at the ED, which in turn leads to unnecessary ED overcrowding. Moreover, a prolonged LOS at the ED increases the patient’s mortality [30,31,32].

In the conducted study, the median ED length of stay was 1 day and 21 h (the mean time was 4 days and 6 h). It is well-known that the ED LOS depends on the patient’s condition and specific sub-groups of the ED population but even more on the organization of care. An improvement in the consultation process can enhance the flow of patients and, as a result, shorten the ED LOS.

The most important factors that are associated with the hospitalization time of patients with pneumonia include predictors related to acute disease (abnormal blood results, clinical signs of severity, severity markers, development of complications) or comorbidities (alcohol consumption, dysphagia, chronic renal failure, neoplastic disease, urinary catheterization, secondary urinary tract infection) and other factors that focus on the social situation of the patient (assessment of the family caregiver’s involvement, active and early involvement of the family in the discharge process, effective communication with the family) [30].

According to Garau and co-workers, PSI high-risk classes (IV, V), positive blood culture results in microbiological testing, ICU admission, multilobar involvement, and alcohol consumption were associated independently with the prolonged LOS. In the study conducted by Garoua and co-workers (3233 patients with CAP, mean age: 66.6 ± 18.5 years; range: 18–100 years, admitted to Spanish university tertiary-care hospitals), the mean LOS was 11.5 days (range: 1–111 days), with the median LOS of 9 days (interquartile range: 7–14 days) [33].

The optimal hospital LOS is difficult to define and varies markedly among centres. In our study, the average hospitalization time was 15 days and 7 h (median: 12 days). In the multicentre study of 875 patients conducted by Suter-Widmer and co-workers, the mean LOS was 9.8 days. The authors revealed that older age, respiratory rate > 20 pm, nursing home residence, chronic pulmonary disease, diabetes mellitus, multilobar CAP, and pneumonia severity index class were independently associated with longer hospitalization [30].

In our observation, longer hospitalization time might be associated with the fact that we had nearly four times more patients with comorbidities. More than 65% of patients had at least one comorbidity versus 16% of patients with coexisting illnesses, as reported by Suter-Widmer et al., who analysed 875 patients with community-acquired pneumonia in Switzerland [30]. The LOS depended not only on the resolution of acute disease but also on organizational issues.

In the studied group of patients, the three most commonly co-occurring disorders included hypertension, renal and heart diseases, and diabetes mellitus. Almost 35% of the cohort had only one comorbid disease, while two comorbidities were present in 17% of patients with pneumonia. Sorino and co-workers found that chronic kidney disease and end-stage renal disease increase the risk of pneumonia [34]. Smoking, alcoholism, compromised immune status, chronic obstructive pulmonary disease (COPD), cardiovascular disease, cerebrovascular disease, chronic liver or renal disease, diabetes mellitus, and dementia have also been mentioned in research studies [7,35]. In the international multicentre observational point-prevalence study of adult patients hospitalized for CAP in 54 countries worldwide, which was conducted by Di Pasquale et al., up to 18% of hospitalized patients had at least one risk factor for immunosuppression. Risk factors for immunosuppression were independently associated neither with *Pseudomonas aeruginosa* nor with fungal infections and viral infections other than influenza [36].

Pneumonia can be caused by various microorganisms: bacteria, viruses, and fungi. Textbooks and elaborations still state that the most common bacterial aetiological factors include *Streptococcus pneumoniae* and type b *Haemophilus influenzae* (Hib) [8,14,29,37,38,39]. The aetiological factor of pneumonia is not always identified. The clinical picture and radiological examination do not allow pathogens to be identified. Microbiological testing is not always done. The IDSA/ATS recommendations for microbiological testing in adults with suspected community-acquired pneumonia without any innate and acquired immunity impairment are as follows:Routine microbiological testing of sputum is not recommended in outpatients;Lower respiratory tract samples for microbiological testing should be collected from inpatients:
a.With severe pneumonia before antimicrobial therapy is started, who were treated for methicillin resistant *Staphylococcus aureus* (MRSA) or *Pseudomonas aeruginosa* infection,b.Previously treated for MRSA or *P. aeruginosa* infection, mostly due to respiratory tract infection, andc.Who were hospitalized or treated with parenteral antibiotics within the past 90 days [14,19].

According to the National Health Statistics Reports based on the follow-up at 94 hospitals in the U.S., microbiological testing was performed in 31.2% of patients at the ED and in 11.5% of CTs [40]. Biological material from the lower respiratory tract was obtained from 8.3% of patients, with blood (48.7%) and urine (35.9%) collected more often for microbiological testing. In the observation carried out, community-acquired pneumonia (CAP) could be suspected in the majority of patients; however, taking into account the characteristics of those patients (with transplant history, cancer and renal diseases, requiring haemodialysis, advanced age), it can be recognized that typical aetiological factors involving *Streptococcus pneumoniae*, *Haemophilus influenzae,* and *Staphylococcus aureus* infections are not necessarily dominant in this group. Bacteria and fungi were cultured from 25 (75.7%) respiratory tract samples. The presence of fungi (*Candida* spp. and *Aspergillus* spp.) was detected in 47.2% of samples. In the study conducted by Gajewska M. et al., who used national public statistical data from the National Institute of Public Health—National Institute of Hygiene to monitor 1,038,810 pneumonia hospitalisations in the period between 2009 and 2016 in Poland, the most commonly recorded cause of pneumonia according to the ICD-10 code was J18—pneumonia, organism unspecified (190.4–220.6/100,000). Bacterial pneumonia, not elsewhere classified (J15) and viral pneumonia, not elsewhere classified (J12) were recognised in 75.4–92.2/100,000 and 22.7–37.5/100,000 of the population, respectively [29]. In our observation, *Hemophilus influenza* was cultured from the respiratory tract of one patient only. *Streptococcus pneumoniae* was identified in blood samples obtained from two patients (negative result of the lower respiratory tract culture). We draw attention to the incidence of fungi cultured from the upper respiratory tract. Fungi (most commonly *Candida spp.*, 44%) were considered to be the aetiological factor based on microbiological testing of material from the lower respiratory tract in almost 47% of patients. Invasive fungal infections most commonly affect immunosuppressed patients, who constituted a significant percentage of our cohort. Correct interpretation of the microbiological test result is difficult or even impossible sometimes, in particular if *Candida* spp. belonging to the microbiome of the upper respiratory and digestive tract mucosa are identified [41]. *Pseudomonas aeruginosa* (8.3%) and *Klebsiella pneumoniae* (8.3%) were relatively often identified. Those bacteria are the most common cause of hospital-acquired infections as well as aspiration pneumonia, which develops as a result of aspiration of contents from the upper respiratory or digestive tracts and occurs most frequently in the elderly with alcoholism or after endoscopic procedures, who were members of our study group. Because of the small number of microbiological tests conducted, the actual aetiological agents of pneumonia in our study are largely speculative. We believe that it is reasonable to have microbiological testing performed in order to thoroughly understand the profile of patients presenting to the ED of our hospital. Local microbiological epidemiology and drug-resistance profile are critical for successful empirical antibiotic therapy [42,43].

Lower respiratory tract infections are not only one of the most frequent reasons for consultation as part of the primary care and at the ED, but they are also a cause of high antimicrobial-prescription rate [37]. Empirical treatment depends on the aetiology of infection influenced by the following factors: comorbidity, baseline functional status, severity of pneumonia, previously received antimicrobial drugs, contact with the hospital system, or even place of residence [37]. Generally, therapy should be initiated as soon as possible.

In the conducted study, almost 92% of patients received an antimicrobial drug, with the vast majority receiving it during the ED stay. Antimicrobial drugs were prescribed in accordance with the national guidelines [12], which are in line with the ATS/IDSA Clinical Guideline on Community-Acquired Pneumonia [14]. In our study, patients received an antibiotic after an average of almost 42 h from ED admission (median time: almost 6 h). Nearly 70% of patients received an antimicrobial therapy within ≤ 1 h of being admitted to the ED. It can be assumed that the short time from reporting to the ED to the first dose of antibiotic is beneficial for patients with suspected community-acquired pneumonia and correlates with a favourable outcome. This correlation could not be clearly established even in studies involving large groups of patients. In a study of 13,771 patients (≥ 65 years of age), Houck at al. reported that administering an antibiotic within 4 h of ED arrival led to a 15% reduction in 30-day mortality [44]. Conversely, a study of 16,313 adults found that administering antibiotics within 4 h did not benefit short-term mortality [45]. Current treatment recommendations for patients with CAP do not assign a specific period for antibiotics administration [46].

It is important to collect samples for microbiological testing before administering anti-infective drugs.

This observational study is the first study conducted as part of the PEDRO project. It has been initiated so that emergency physicians can heuristically use the best available evidence in a local setting with pragmatic interventions to improve critical outcomes in patients with severe CAP.

## Figures and Tables

**Figure 1 pathogens-11-00779-f001:**
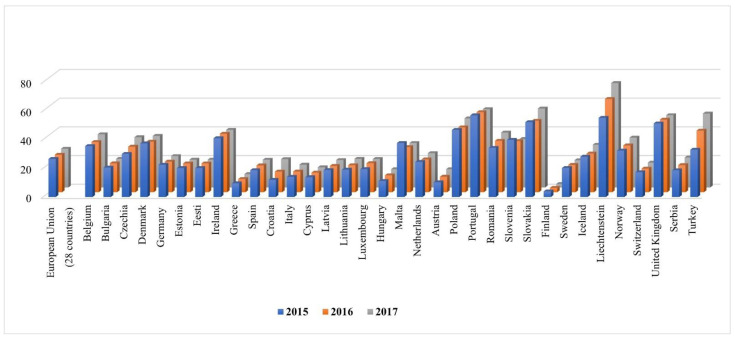
Deaths from pneumonia in EU region.

**Table 1 pathogens-11-00779-t001:** Demographics and history of patients visiting the ED who were diagnosed with pneumonia, intra-hospital movement and in-hospital mortality rates.

Feature	Statistic Level	2019		2020		Total	
	N	184		214		398	
Patient age	Range	21–98		19–100		19–100	
	Mean	70.85		70.93		70.89	
	Median	73		72,5		73	
	SD	17.82		17.11		17.42	
		N	%	N	%	N	%
Age groups (years)	18–65	62	33.7	74	34.6	136	34.2
	66–80	59	32.1	68	31.8	127	31.9
	>80	63	34.2	72	33.6	135	33.9
Sex	Female	66	35.9	80	37.4	146	36.7
	Male	118	64.1	134	62.6	252	63.3
Transport by EMT	Yes	117	63.9	161	75.2	278	70.2
	No	66	36.1	52	24.3	118	29.8
**History of patients visiting the ED**							
Patients visiting the ED with severe pneumonia		184	100	214	100	398	100
Hospitalized patients		184	100	201	93.9	385	96.7
Patients not admitted to the ED (transferred to other hospital * or self-dismissal)		0	0	13	6.1	13	3.3
Patients admitted to the ICU		21	11.4	24	11.2	45	11.3
Patient admitted to internal medicine units		163	88.6	177	82.7	340	85.4
Patients who died (at the ED)		0	0	1	0.47	1	0.25
Patients who died (at the ICU)		10	5.4	13	6.1	23	5.8
Patients who died (at the internal medicine unit)		17	9.2	23	10.7	40	10.0
Patients who died (total)		27	14.7	37	17.3	64	16.1

N, number of patients; SD, Standard deviation; EMT, emergency medical team; ICU, intensive care unit; * COVID-19 units or Institute of Tuberculosis and Lung Diseases.

**Table 2 pathogens-11-00779-t002:** Triage patient characteristics.

Triage	Statistic Level	2019	2020	Total
HR (beats/min)	Number of patients	161	205	366
	Range	30–200	60–150	30–200
	Mean	91.6	91.4	91.5
	Median	90	90	90
	SD	21.0	16.6	18.7
SP (mmHg)	Number of patients	157	205	362
	Range	15–209	13–202	13–209
	Mean	122.9	125.33	124.3
	Median	122	125	124.5
	SD	30.68	27.6	28.9
DP (mmHg)	Number of patients	157	205	362
	Range	30-110	30–120	30–120
	Mean	74.3	74.83	74.6
	Median	78	78	78
	SD	15.4	13.52	14.3
T (°C)	Number of patients	118	188	306
	Range	36–98	35.9–40.1	35.9–98
	Mean	38.5	37.2	37.7
	Median	37	36.8	37
	SD	7.7	0.9	4.9
OS (%)	Number of patients	153	200	353
	Range	15–100	58–100	15–100
	Mean	91.3	92.2	91.8
	Median	95	95	95
	SD	11.4	7.0	9.2
GCS	Number of patients	148	201	349
	Range	3–15	3–15	3–15
	Mean	14.2	13.9	14.0
	Median	15	15	15
	SD	1.5	1.7	1.6

SD, Standard deviation.

**Table 3 pathogens-11-00779-t003:** Symptoms, clinical diagnosis, and comorbidities in patients visiting the ED who were diagnosed with pneumonia.

Presenting Symptom	2019		2020		Total	
	N 184	%	N 214	%	N 398	%
Dyspnoea	69	37.5	101	47.2	170	42.7
Malaise	24	13	101	47.2	125	31.4
Fever/feeling of fever	26	14.1	56	26.2	82	20.6
Cough	5	2.7	24	11.2	29	7.3
Pleuritic chest pain	5	2.7	22	10.3	27	6.8
Urinary tract infection	24	13	2	0.9	26	6.5
Other: sepsis, acute respiratory failure, taste/smell disturbances, dementia	18	9.8	9	4.2	27	6.8
**Number of reported symptoms**						
One symptom	145	78.8	86	40.2	231	58.0
Two symptoms	11	6.0	88	41.1	99	24.9
Three symptoms	1	0.5	13	6.1	14	3.5
Four symptoms	0	0.0	1	0.5	1	0.3
Five symptoms	0	0.0	2	0.9	2	0.5
**Clinical diagnosis**						
Pneumonia	181	98.4	210	98.1	391	98%
Infection in COPD	15	8.2	22	10.3	37	9.3
Aspiration pneumonia	5	2.7	11	5.1	16	4.02
Other: sepsis, urinary tract infection, COVID-19, tuberculosis	0	0	9	4.2	9	2.3
**Comorbidities/co-occurring conditions**						
Hypertension	7	3.8	55	25.7	62	15.6
Renal disease	15	8.2	47	22.0	62	15.6
Heart disease	3	1.6	52	24.3	55	13.8
Diabetes mellitus	16	8.7	37	17.3	53	13.3
Oncological history	16	8.7	32	15	48	12.1
Solid organ transplantation	24	13	22	10.3	46	11.6
Mental disorders	12	6.5	29	13.6	41	10.3
Bronchial asthma/COPD/emphysema	0	0	26	12.1	26	6.5
Urinary tract infection	0	0	25	11.7	25	6.3
Alcohol dependence	3	1.6	16	7.5	19	4.8
**Number of comorbidities**						
1 comorbidity	71	38.6	67	31.3	138	34.7
2 comorbidities	11	6.0	57	26.6	68	17.1
3 comorbidities	1	0.5	28	13.1	29	7.3
4 comorbidities	0	0.0	17	7.9	17	4.3
5 comorbidities	0	0.0	5	2.3	5	1.3
6 comorbidities	0	0.0	1	0.5	1	0.3

N, number of patients; COPD, chronic obstructive pulmonary disease.

**Table 4 pathogens-11-00779-t004:** The ED length of stay and hospitalization duration.

Characteristic	Statistic Level	2019	2020	Total
ED length of stay (days)	N	182	208	390
	Mean	4.26	2.95	3.56
	Median	1.35	1.99	1.88
	SD	22.95	3.45	15.87
Hospital length of stay (days)	N	177	189	366
	Mean	15.62	16.01	15.82
	Median	11	13	12
	SD	26.63	11.56	20.27

N, number of patients; SD, Standard deviation.

**Table 5 pathogens-11-00779-t005:** Time from ED admission to antibiotic administration.

Characteristic	Statistic Level	2019	2020	Total
Time from ED admission to antibiotic administration (hours)	N	171	176	347
	Mean	68.38	18.23	41.28
	Median	5.22	6.48	5.81
	SD	698.32	61.37	475.38

N, number of patients; SD, Standard deviation.

**Table 6 pathogens-11-00779-t006:** Antimicrobial treatment of inpatients with pneumonia.

Antimicrobial Drugs		N (365)	%
**Β-lactams**	Ceftriaxone	283	77.5
	Meropenem	36	9.9
	Amoxicillin and clavulanic acid	16	4.4
	Piperacillin and tazobactam	10	2.7
Ciprofloxacin		78	21.4
Clarithromycin		22	6.0
Levofloxacin		11	3.0
Metronidazole		16	4.4
Vancomycin		9	2.5
Clindamycin		2	0.55
Sulfamethoxazole + trimethoprim Linezolid		3 1	0.8 0.3
Tigecycline		1	0.3
Antifungal drug			
Fluconazole, Caspofungin Pentamidine		8 1	2.2 0.3
Antiviral drug (oseltamivir, ganciclovir)		4	1.1

N, number of patients.

**Table 7 pathogens-11-00779-t007:** The number of performed chest X-ray and chest CT tests.

Procedures	2019 (N 184)		2020 (N 214)		Total (N 368)	
	N	%	N	%	N	%
Chest X-ray	171	92.8	76	35.5	247	62.1
Chest CT	81	44	160	74.8	241	60.6
Chest X-ray and CT	70	38	34	15.9	103	25.9

N, number of patients.

**Table 8 pathogens-11-00779-t008:** Microbial species cultured from blood, lower respiratory tract, and urine samples.

Group of Bacteria and Species		Blood Sample		Respiratory Specimens		Urine Sample	
		N	%	N	%	N	%
**Gram-positive bacteria**							
*Staphylococcus*	*Staphylococcus aureus*	5	9.8	2	5.5	1	2
	*Staphylococcus epidermidis*	7	13.7	-	-	-	-
	*Staphylococcus hominis*	4	7.8	-	-	-	-
	*Staphylococcus haemolyticus*	1	2	-	-	-	-
	*Staphylococcus salivarius*	-	-	1	2.8	-	-
	*Staphylococcus pettenkoferi*	2	3.9	-	-		
*Streptococcus*	*Streptococcus pneumoniae*	2	3.9	-	-	-	-
	*Streptococcus agalactiae*	-	-	1	2.8	-	-
	*Streptococcus oralis*	1	2	-	-	-	-
	*Staphylococcus saccharolyticus*	1	2	-	-	-	-
	*Streptococcus parasanguinis*	-	-	1	2.8	-	-
*Enterococcus*	*Enterococcus faecalis*	3	3.9	1	2.8	6	12
	*Enterococcus faecium*	1	2	2	5.5	1	2
Other Gram-positive cocci	*Rothia mucilaginosa*	-	-	2	5.5	-	-
	*Finegoldia magna*	-	-	-	-	1	2
Gram-positive rods	*Corynebacterium tuberculostearicum*	1	2	-	-	-	-
	*Clostridium perfringens*	1	2	-	-	-	-
**Gram-negative bacteria**							
Gram-negative rods	*Enterobacter cloacae*	3	3.9	-	-	-	-
	*Escherichia coli*	6	11.8	1	2.8	18	36
	*Stenotrophomonas maltophilia*	-	-	1	2.8	-	-
	*Pseudomonas aeruginosa*	3	3.9	3	8.3	7	14
	*Citrobacter freundii*	-	-	-	-	1	2
	*Klebsiella pneumoniae*	3	3.9	3	8.3	9	18
	*Pseudomonas stutzeri*	1	2	-	-	-	-
	*Proteus mirabilis*	-	-	-	-	1	2
	*Klebsiella oxytoca*	-	-	-	-	1	2
	*Haemophilus influenzae*	-	-	1	2.8	-	-
Fungi	*Candida* spp. *(C. albicans*, *C. glabrata, C. dubliniensis*, *C. tropicalis)*	* 6	11.8	16	44.4	4	8
	*Aspergillus spp.*	-	-	1	2.8	-	-
total		51		36		50	

N, number of patients; * antigen detection.

**Table 9 pathogens-11-00779-t009:** The influence of symptoms and comorbidities on the mortality rate.

	IN	Yes	IND (64)	chi-Square	df	*p*
**Presenting symptoms**						
Dyspnoea	155	20	44	3.3111	1	0.069
Malaise	108	23	41	0.239	1	0.625
Fever/feeling of fever	68	4	60	8.559	1	0.003
**Comorbidities**						
Hypertension	56	8	56	0.168	1	0.682
Renal disease	41	8	56	0.089	1	0.799
Heart disease	53	13	51	0.794	1	0.373
Diabetes mellitus	48	7	57	1.489	1	0.222
Oncological history	57	14	49	8.102	1	0.004 *
Solid organ transplantation	46	2	62	10.884	1	0.001 *
Mental disorders	37	12	52	5.846	1	0.016 *

N, number of patients; IN, total number of incidence; IN, number of incidence in individuals who died; * statistically significant.

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
