# Peer review of "Change for the Better: Severe Pneumonia at the Emergency Department"

_pathogens, 2022, doi:10.3390/pathogens11070779_

Round 1

Reviewer 1 Report

Well written paper, very interesting topic with important clinical implications 

Author Response

Thank you for review and recommending the publication.

Best regards.

Reviewer 2 Report

The present study entitled "Change for the better: Severe pneumonia at the Emergency Department" aims to evaluate demographic features, clinical patterns and 72 history of hospitalization associated with patients who suffer from pneumonia requiring 73 hospitalization and their impact on mortality".

I consider the article of great interest for this magazine, I only recommend that the authors approach the topic in the same way from the perspective of COVID-19, which would make their work even more interesting for readers.

I suggest including the following reference as part of your introduction: doi: 10.24875/RIC.20000207. In this work, the characteristics of patients with COVID-19 pneumonia as well as the factors associated with mortality are reported for the first time.

Likewise, I recommend authors to include the following reference in their discussion: doi: 10.17179/excli2021-3413. That would allow discussing the effects of the various treatments in patients with COVID-19 pneumonia, and their association with various outcomes.

Once these changes are added, I recommend that the article can be published.

Author Response

We thank the Reviewer for comments and for recommending the publication,

The publications indicated have just been used to present our further study on patients with Covid-19. The observations described in this manuscript were conducted through October 15, 2020; after 15 October 2020 only patients with confirmed Covid-19 were admitted to our hospital's ED therefore, the study was terminated, and we started a study with a cohort of patients infected with SARS-CoV-2. We are currently preparing a manuscript.

Reviewer 3 Report

In this manuscript, the authors have presented a detailed observational study of the demographics, signs and symptoms, co-morbidities, length of hospital stay, diagnosis, management and mortality of adults presenting with severe pneumonia at a single-center Emergency Department in Warsaw, Poland between1 January 2019 and 15 October 2020.

This study provides important empirical data to support national recommendations in Poland for the management of severe pneumonia and patient care, which need to be urgently updated, as the authors emphasize, due to changing variables such as population demographics, infectious agents, methods of diagnosis and hospital cost policy. The authors also highlight that the mortality rate from pneumonia is higher in Poland compared with other European countries.

I would recommend the following amendments, which I feel would improve the overall quality of the manuscript:

i) In Table 5 in the Results, it would be better to present the range of time from ED admission to antibiotic administration, instead of the mean. Also, a comment about the variability in the time between ED admission and antibiotic administration would be helpful to include in the Discussion and how the findings in this study compare with those observed in other countries.

ii) It would be helpful to the reader to add a summary/conclusion paragraph to the Discussion, which comments on the implications and importance of the study findings, e.g. did the rate of deaths from pneumonia support the findings in Figure 1 of a higher rate in Poland compared with other EU countries? How did the diagnosis and management of severe pneumonia compare? What recommendations could be made to national policy? Also, adding some future directions for the research would be useful, e.g. is there a plan to expand the observations to other Polish ED departments?

iii) Similarly, it would be also be useful to add a conclusion sentence(s) to the Abstract to highlight the importance of the study findings to the reader.

iv) Please check the spellings of authors listed in the References, e.g. in reference #3, “Shibet” should be “Shiber”.

Author Response

We would like to thank for the time dedicated to our work and the improvement of the manuscript we have been able to make with the help of the comments provided. Please see the attachment.

Best regards
